# A Statistical Evaluation to Compare and Analyze Estimations of the Diffusion Coefficient of Pertechnetate ($^{99}$TcO$_{4-}$) in Compacted Bentonite

**Chuan-Pin Lee** [1,†], **Yanqin Hu** [1,†], **Dongyang Chen** [1], **Neng-Chuan Tien** [2,*], **Shih-Chin Tsai** [2,*], **Yunfeng Shi** [3,4], **I-Hsien Lee** [5] **and Chuen-Fa Ni** [5,6]

1    School of Nuclear Sciences and Engineering, East China University of Technology, Nanchang 330013, China; bennis6723@139.com (C.-P.L.); hyqdrx@163.com (Y.H.); chen_dongyang2021@163.com (D.C.)
2    Nuclear Science and Technology Development Center, National Tsing Hua University, Hsinchu 30013, Taiwan
3    State Key Laboratory of Nuclear Resources and Environment, East China University of Technology, Nanchang 330013, China; syf541006935@126.com
4    Department of Nuclear Environmental Science, China Institute for Radiation Protection (CIRP), Taiyuan 030006, China
5    Center for Environmental Studies, National Central University, Taoyuan City 32001, Taiwan; sfdff327@gmail.com (I.-H.L.); nichuenf@gmail.com (C.-F.N.)
6    Graduate Institute of Applied Geology, National Central University, Taoyuan City 32001, Taiwan
*    Correspondence: nctien@mx.nthu.edu.tw (N.-C.T.); sctsai@mx.nthu.edu.tw (S.-C.T.); Tel.: +86-18258244042 (N.-C.T.)
†    These authors contributed equally to this work.

**Abstract:** Various numerical methods have been being extensively used to estimate the diffusion parameters of pertechnetate ($^{99}$TcO$_{4-}$) in compacted MX80 bentonite clay using through-diffusion (TD) techniques. In this study, an improved TD column test was applied, and the diffusion fluxes of tritium (HTO) as a non-reactive radionuclide, and $^{99}$TcO$_{4-}$ with various dry densities, were compared under the same experimental conditions. Similar results were obtained for the apparent diffusion coefficients of HTO and $^{99}$TcO$_{4-}$ using three estimation methods: a graphical method applying the asymptote calculation, an analytical solution using Lsqcurvefit installed in MATLAB, and the Marquardt–Levenberg optimization algorithm in the HYDRUS-1D inverse method. The statistical analysis showed that the densities using a one-way analysis of variance (ANOVA) for the three estimation methods ranged from 1200 to 2000 kg/m$^3$, which indicated that there were no obvious differences between HTO and $^{99}$TcO$_{4-}$. In general, the diffusion parameters of $^{99}$TcO$_{4-}$ were lower than those for HTO due to anion exclusion effects and lower accessible porosity.

**Keywords:** diffusion; MX80 bentonite; HYDRUS-1D; asymptote; ANOVA

## 1. Introduction

With almost 70% of the world's operating nuclear reactors now over 30 years of age, countries around the world are assessing whether to allow reactor operations to continue past the 50–60 year mark and potentially up to 80 years. Ensuring a proper legal framework for the long-term operation of nuclear power reactors is a key component of such considerations. There exists a strong international consensus that geologic repositories for the disposal of spent nuclear fuel (SNF) and high-level radioactive waste (HLW) provide the greatest protection in terms of public health and the environment (U.S. Nuclear Waste Technical Review Board, 2016) [1]. In fact, the safe disposal of SNF and HLW has been the most controversial issue associated with nuclear energy around the world in the past decades. Recently, Sweden, Finland, and China embraced the geological disposal of HLW or SNF. Compacted bentonite in radioactive waste repositories is widely used as a buffer/backfill material in engineered multi-barrier system construction. Moreover,

understanding the diffusion transport process through compacted bentonite is crucial to assess the performance of these barriers [2–4]. The diffusion coefficients are therefore key parameters for predicting radionuclide migration behavior through the engineering barriers and can be applied in the performance and safety assessment of engineered barrier systems. MX-80 is a type of commercial bentonite that is generally utilized as an engineering barrier material and can be chosen as a potential buffer and backfill material for the geological disposal of spent nuclear fuel in several countries due to its excellent swelling ability, good sorption properties, low permeability, high thermal conductivity, and the highly effective characteristics of the exchangeable cations [5,6].

Technium-99 ($^{99}$Tc) is a pure β-emitting radioisotope with a maximum particle energy of 294 keV and a key radionuclide present in nuclear reactors from $^{235}$U and $^{239}$Pu at a maximum fission yield of 6.06%. The threats imposed by the high mobility of $^{99}$Tc under geo-environmental oxidizing conditions [7,8] and the fact that it is rather long-lived ($2.13 \times 10^5$ y) point out the importance of accurately assessing the safety of deep geological facilities. Similar to the SNF disposal program in Sweden, the KBS-3 concept with the multiple barrier principle [4] has also been adopted in various national projects. In an oxidizing aqueous environment, $^{99}$Tc exists in bentonite predominantly in the form of anions. Laboratory diffusion studies and proper theoretical models are required for the prediction of diffusion flow in field applications. Several laboratory diffusion studies have been conducted for the purpose of estimating the mass transport parameters. Numerous studies have investigated the sorption and/or diffusion characteristics of $^{99}$Tc (or analogous Rhenium; Re) on bentonite [9–16].

Diffusion transport studies through compacted bentonites are important in the areas of landfill construction, nuclear waste repositories, and soft ground remediation. The through-diffusion (TD) technique is a traditional method used to obtain the diffusion coefficient of weak and non-sorbing tracers [17]. In this paper, we adopted the TD method with the sample located between two chambers labeled "in reservoir" and "out reservoir". The cumulative concentration method developed by Crank (1975) [18] is often employed using the asymptote calculation to obtain the diffusion parameters, including the apparent diffusion coefficient (Da) and the effective diffusion coefficient (De). However, a significant ambiguity in the choice of the numerical model or simulation often makes an accurate analysis of the diffusion transport difficult. Recently, there have been several numerical analysis or methods used to obtain diffusion parameters, including ANADIFF [19,20], quantum-behaved particle swarm optimization (QPSO) [21,22], fitting for diffusion parameters (FDP) [23,24], the dual period diffusion model (DPDM) [25], and a parameter identification process based on an iterative and analytical method (PIPIAM) [26] etc., and some commercial software programs (i.e., HYDRUS) used for groundwater transport simulations are also convenient tools that can be used to estimate the diffusion coefficients based on the experimental results of TD techniques [27]. In fact, there are always experimental errors and uncertainty associated with each type of numerical analysis or method, so it is necessary to summarize or determine a suitable way to examine radionuclide diffusion behavior.

Numerical and estimation analyses of the diffusion parameters from compacted MX-80 bentonite as well as their significance and influence based on a statistical analysis are compared and discussed in this work. The aim of this paper was first to conduct through-diffusion (TD) experiments for HTO and $^{99}$TcO$_{4-}$ in compacted MX-80 bentonite with a wide range of densities using the through-diffusion technique. In this work, a statistical evaluation of various numerical estimations of diffusion parameters is described and investigated based on the HTO and $^{99}$TcO$_{4-}$ TD experimental results. A secondary objective was to compare and estimate the diffusion coefficients using various numerical estimations, including the graphical asymptote calculation, the Lsqcurvefit (non-linear Least Squares Curve Fit) analysis by MATLAB, and the HYDRUS-1D inversing calculation. Finally, a statistical evaluation using a one-way analysis of variance (ANOVA) was applied to determine the differences in HTO and $^{99}$TcO$_{4-}$ diffusion in compacted MX-80 bentonite.

## 2. Experimental Procedure

### 2.1. Bentonite Clay Sample, Radioisotopes Standards, Chemical Reagents, and the Liquid Phase

Commercial and fine granular sodium (Na) types of Wyoming bentonite, MX-80 were purchased from the American Colloid Company (AMCOL, Hoffman Estates, IL, USA). The samples had an average particle size ranging between 75 and 100 μm and a true density (t) of 2700 kg/m$^3$ [5,6]. The chemical composition of the simulated underground water (SGW) is listed in previous works [26,27], which comprised typical water samples obtained at a depth of 300–500 m from six boreholes drilled through crystalline rock [2]. The tracer standards were traceable to Packard Spec-ChecTM (Packard Machinery Co, Westford, MA, USA) and NIST SRM 4288A (NIST, Gaithersburg, MD, USA) for HTO and $^{99}$TcO$_{4-}$, respectively. The collected samples were measured with a liquid scintillation analyzer (LSC, Packard 3170 AB/TR, Shelton, CT, USA) with a counting efficiency of around 91% for $^{99}$Tc, and the scintillation cocktails for HTO and $^{99}$Tc were Gold$^{TM}$ LLT (PerkinElmer, Waltham, MA, USA) and Ultima Gold$^{TM}$ AB (PerkinElmer, Waltham, MA, USA), respectively. The mixing process involved placing 10 mL of the sample in a 20-mL polyethylene counting vial to which 10 mL of the scintillation cocktail was added. All chemical reagents used were of analytical purity, and the resistivity of Milli-Q$^®$ ultrapure water was 18.2 MΩ.cm in the experiments.

### 2.2. Physical Characterization of Bentonite Clay

The structural analysis of bentonite clay was performed using a Fourier transform infrared (FT-IR) spectrometer (Horiba FT-730 spectrometer, Kyoto, Japan). The spectra data were recorded in a range from 4000 to 400 cm$^{-1}$ with samples pressed with KBr powder. A wavelength dispersive X-ray (WDX) fluorescence spectrometer, AXIOS (Panalytical, Malvern, UK), was employed for the semi-quantitative analysis of the chemical composition, and the specific surface area was determined using N$_2$-BET measurements (Micromeritics ASAP 2020, Micromeritics, Norcross, GA, USA). The mineralogical analysis was carried out on the powder samples using a Bruker-D8 Advance XRD instrument with Cu K$α$1 radiation (wavelength = 1.5418 Å) operating with the following operational parameters: a 40 kV voltage and a 40 mA current, a 0.01 step size, a 1°/min speed, and a scanning angle 2θ ranging from 5 to 80. The phase identification, quantitative analysis, and structural determinations were compared with the JCPDS (Joint Committee on Powder Diffraction Standards) in the PDF02 (Powder Diffraction File 02, ICDD, Newtown Square, PA, USA) software database.

### 2.3. Experimental Set-Up for Through-Diffusion with a Constant Inlet Concentration–Constant Outlet Concentration

The amount of MX-80 bentonite, weighed based on the chosen dry densities (1200, 1400, 1600, 1800, and 2000 kg/m$^3$), was pressed into the sample holder using a manual compressor. The sample was 0.3 cm thick and the cross-sectional area for the bentonite discs measured 19.6 cm$^2$. The sample slab was inserted in a sample holder and was sandwiched between two cells with a volume ranging between approximately 80 and 100 mL. The diffusion cells were made of polypropylene (PP). Figure 1 shows a schematic description of the diffusion cell for the compacted bentonite used in the through-diffusion experiments. A Teflon O-ring (5 cm in O.D.—outer diameter, and 3.6 cm in I.D.—inner diameter, 2.0 cm thick) and porous constrainer (5 cm in diameter and 0.7 cm thick) made of reinforced PP were used to reduce cell deformation, minimize diffusion resistance, and resist the swelling pressure induced by the bentonite clay following saturation with water. A commercially available Amer-Sil membrane, a type of microporous polyvinyl chloride (PVC)/silica separator exhibiting high enough tensile strength to avoid swelling deformation of the compacted bentonite and the rupture of the membrane filter during water saturation was used in this study to confine the sample and also to prevent the release of bentonite particles [28].

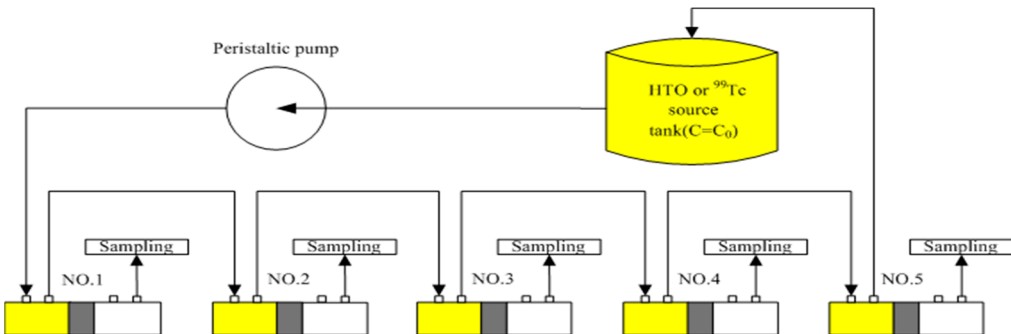

**Figure 1.** Experimental apparatus showing five through-diffusion cells using bentonite compacted to various dry densities (1200, 1400, 1600, 1800, and 2000 kg/m$^3$) [26,27].

Before the through-diffusion experiments, the compacted bentonite was saturated with SGW with an air pump at a suction pressure of 0.2–0.3 kPa to accelerate saturation for about two weeks. At the start of the experiment, a small amount of HTO was added into the inlet (source) reservoir until an activity concentration of approximately 5 Bq/mL ($C_0^{HTO}$) was reached, and the outlet reservoir was initially filled with fresh SGW. The diffusion cells (columns) were placed in a horizontal direction during the diffusion experiment to avoid advection or dispersion through the sample owing to pressure differences. All of the liquid solution at the outlet (measurement) reservoir was totally removed and was collected daily. A total of 10 mL of this liquid was sampled and monitored for the determination of HTO radioactivity using an LSC. After sampling, a period of half an hour for the circulation of the source tank (5 L) with a peristaltic pump was needed. Following the replacement of HTO in the inlet reservoir with SGW spiked with the $^{99}$Tc radionuclide (tracer), the activity concentration of $^{99}$Tc reached roughly 20 Bq/mL ($C_0^{Tc}$), and then the sampling and measurement procedures were repeated. Figure 1 also illustrates that the experimental setup was the same as that for HTO, including the five diffusion cells with various dry densities (1200, 1400, 1600, 1800, and 2000 kg/m$^3$).

The activity of the diffusing solute in the source reservoir remained almost constant ($C_0$) during the entire experimental period, and it remained at almost zero or close to zero (and thus constant) in the outlet reservoir. If the steady state was reached, the diffusive flux of radionuclides (HTO or $^{99}$TcO$_{4-}$) across the sample was constant. In this study, a large volume of the sample, 5 L, with high radioactivity was used in the source tank connected to each inlet reservoir to minimize decreases in the activity concentration, and frequent sampling was carried out as well as daily refreshing of the SGW in the outlet reservoir, which made it possible to maintain the concentration at a near constant level (not varying more than 5%). The through-diffusion curve was obtained by adding up the activity concentrations of either HTO or $^{99}$TcO$_{4-}$ in the outlet reservoir and plotting the cumulative radioactivity ratio CR(t) versus the elapsed time t.

*2.4. Statistical Analysis Using the Various Numerical Estimations of the Diffusion Parameter*

Both the initial and boundary conditions limit the through-diffusion method. The diffusion parameters, including the apparent diffusion coefficients (Da) and capacity factor ($\alpha$), were obtained using the mathematical and analytical solutions for the concentration profile $C(x,t)$ in the compacted bentonite at time ($t$), as given by Crank (1975) [18]:

$$\frac{C(x,t)}{C_0} = 1 - \frac{x}{L} - \frac{2}{\pi} \sum_{n=1}^{\infty} \frac{1}{n} \sin\left[\frac{n\pi x}{L}\right] \exp\left[\frac{-n^2\pi^2 D_a t}{L^2}\right] \tag{1}$$

The formula of the cumulative concentration ratio (CR(t)) of radionuclide (HTO or Tc) in the measurement reservoir is defined by:

$$\frac{\sum C(t)}{C_0} = \frac{\alpha LS}{V}\left(\frac{D_a t}{L^2} - \frac{1}{6} - \frac{2}{\pi^2} \sum_{n=1}^{\infty} \frac{(-1)^n}{n^2} \exp\left[\frac{-n^2\pi^2 D_a t}{L^2}\right]\right) \tag{2}$$

where $L$ is the length of compacted bentonite (=0.3 cm); $S$ is the cross-section area of the compacted simples (=19.625 cm$^2$), and $V$ is the volume of the outlet (measurement) reservoir (80 to 100 mL).

In this work, there were three numerical estimations, where the graphical asymptote, an analytical solution in MATLAB Lsqcurvefit (The MathWorks, Inc, Natick, MA, USA), and HYDRUS-1D (HYDRUS Engineering Ltd., Gerakas, Greece) were applied to simulate the diffusion of HTO and Tc in compacted MX-80 bentonite with various densities of 1200, 1400, 1600, 1800, and 2000 (kg/m$^3$).

### 2.4.1. Asymptote Estimation

A more convenient and useful method for obtaining the diffusion parameters for (Da and $\alpha$) was applied and fitted using the graphical asymptote (straight line) of HTO and Tc in Equation (2) for sufficient TD experimental time (tf), which indicated that the diffusion process had reached a steady state [29–31].

### 2.4.2. Analytical Estimation Using Lsqcurvefit

A numerical analysis was developed and carried out using a non-linear least squares simulation (the Lsqcurvefit analysis in MATLAB) to obtain the CR(t) of HTO and Tc in this study. The diffusion time required to reach a steady state was compared and discussed [26]. Two different algorithms were implemented: the trust region reflective and Levenberg–Marquardt option, which could be effective and important tools for obtaining Da and $\alpha$ in the future based on the experimental and numerical results of HTO and Tc at the same time.

### 2.4.3. Inverse Estimation

In the TD experiments, the daily activity (HTO or Tc concentration) obtained at the outflow reservoir was used as the input data in HYDRUS-1D for the estimation of Da and $\alpha$ under various bentonite densities [27]. HYDRUS-1D, a Microsoft Windows-based modeling environment, was used to assess efforts to optimize the solute transport parameters of [32–36] in porous media saturated to various degrees. The model was used to solve the one-dimensional Richards' equation for water flow, the Fickian-based advection dispersion equations were used to examine the heat and solute transport, and the Marquardt–Levenberg type parameter estimation technique was used to obtain the selected soil hydraulic and/or solute transport and reaction parameters [37].

### 2.4.4. Statistical Analysis

A one-way analysis of variance (ANOVA) was performed using SPSS 18.0 software package (SPSS Inc., Chicago, IL, USA). A one-way ANOVA combined with the Duncan method was employed to determine the analytical significance ($p$ value) of the three estimations, the various densities, and the radionuclides (HTO and Tc).

## 3. Results

### 3.1. Characterization of the FTIR Analysis

Figure 2 presents the FTIR spectrum of the Na-bentonite, and the absorption band observed at 3634 cm$^{-1}$ (for montmorillonite) corresponded to the stretching vibrations of structural OH groups (OH) in the bentonite. A broad band at approximately 3430 cm$^{-1}$ indicated the H$_2$O-stretching vibrations caused by water molecules in the bentonite. The band at 1643 cm$^{-1}$ was associated with an overtone of the bending vibration of the water.

The characteristic absorption of montmorillonite clay could be observed in the region between 1113 and 1041 cm$^{-1}$, where the strong absorption band at 1047 cm$^{-1}$ was associated with the Si–O bending vibration. Two sharp peaks (795 and 779 cm$^{-1}$) at approximately 800 cm$^{-1}$ were attributed to the presence of quartz (SiO$_2$) [14].

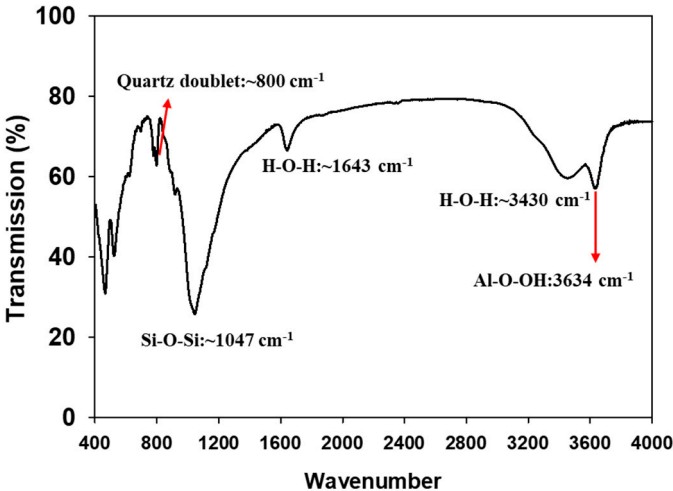

**Figure 2.** FTIR spectrum of MX-80 bentonite.

The N$_2$-BET technique was employed to determine the specific surface area of the radionuclide sorption on the MX-80 bentonite clay, for which the analytical result was approximately 23.0 m$^2$/g. The chemical composition (as determined using XRF) was as follows: 58.9% SiO$_2$, 15.1% Al$_2$O$_3$, 1.6% MgO, 3.1% Fe$_2$O$_3$, 2.2% Na$_2$O, 1.7% CaO, 0.6% K$_2$O, 0.2% TiO$_2$, 0.1% MnO, 0.1% P$_2$O$_5$, and LOI 14.3%. The XRD results revealed that the mineralogical composition of the samples mainly comprised montmorillonite (80%) as well as tridymite (5.1%), plagioclase (4.7%), quartz (3.2%), muscovite (2.5%), gypsum (1.1%), pyrite (0.9%), and illite (0.7%). Pyrite (FeS$_2$) that could potentially retard the transport of radionuclides in some geological materials (host rock or clay) was observed from the XRF and XRD measurements.

### 3.2. Diffusion Behavior of HTO and $^{99}$Tc in the TD Experiments

HTO, a non-reactive tracer, has been shown to reveal the total pore volume of geological media, which is equivalent to the water content. Thus, it is commonly used as a conservative tracer for all chemical elements that are not retained in the solid phase. Table 1 lists the individual values for the apparent diffusion coefficient (Da) and the capacity factor (α) of the radiotracers (HTO and $^{99}$TcO$_{4-}$) in compacted MX-80 bentonite at various densities of 1200, 1400, 1600, 1800, and 2000 (kg/m$^3$) fitted and estimated from three numerical analyses, including the graphical asymptote, the use of Lsqcurvefit to obtain an analytical solution, and HYDRUS-1D in Figure 3. As shown in Figure 3, the diffusion curves generated by CR(t) versus time were consistent, where the breakthrough curves exhibited two phases: (1) a transient phase, where the concentration gradient in the sample built up and (2) a steady state phase, where the concentration gradient in the sample was linear, and the concentration activity in the outlet reservoir increased linearly with time. A summary of the TD experiments on various compacted densities (1200 to 2000 kg/m$^3$) indicated that the time lag required for the HTO and Tc in MX-80 to diffuse out was also approximately 24 h, where the diffusion steady-state was reached in approximate 7 to 10 days due to a constant diffusing flux.

**Table 1.** The through-diffusion conditions for the HTO and Tc in compacted MX-80 bentonite.

| Parameter | Radionuclides (RN) | Method of Estimation | Number of Samples | Bulk Density (kg/m³) | | | | |
|---|---|---|---|---|---|---|---|---|
| | | | | 1200 | 1400 | 1600 | 1800 | 2000 |
| $D_a$ (m²/s) | HTO | Asymptote | 5 | $9.18 \times 10^{-11}$ | $8.01 \times 10^{-11}$ | $3.79 \times 10^{-11}$ | $4.41 \times 10^{-11}$ | $6.60 \times 10^{-11}$ |
| | | Hydrus-1D | 5 | $1.22 \times 10^{-10}$ | $1.13 \times 10^{-10}$ | $7.65 \times 10^{-11}$ | $8.09 \times 10^{-11}$ | $8.55 \times 10^{-11}$ |
| | | Lsqcurvefit | 5 | $1.22 \times 10^{-10}$ | $1.22 \times 10^{-10}$ | $5.76 \times 10^{-11}$ | $6.24 \times 10^{-11}$ | $7.21 \times 10^{-11}$ |
| | | Mean | | $1.12 \times 10^{-10}$ | $1.05 \times 10^{-10}$ | $5.73 \times 10^{-11}$ | $6.25 \times 10^{-11}$ | $7.45 \times 10^{-11}$ |
| | | Standard deviation | | $1.74 \times 10^{-11}$ | $2.21 \times 10^{-11}$ | $1.93 \times 10^{-11}$ | $1.84 \times 10^{-11}$ | $9.98 \times 10^{-12}$ |
| | Tc | Asymptote | 5 | $5.54 \times 10^{-12}$ | $9.94 \times 10^{-12}$ | $1.30 \times 10^{-11}$ | $1.94 \times 10^{-11}$ | $2.05 \times 10^{-11}$ |
| | | Hydrus-1D | 5 | $5.90 \times 10^{-12}$ | $1.13 \times 10^{-11}$ | $1.56 \times 10^{-11}$ | $2.43 \times 10^{-11}$ | $2.60 \times 10^{-11}$ |
| | | Lsqcurvefit | 5 | $1.05 \times 10^{-11}$ | $1.12 \times 10^{-11}$ | $1.32 \times 10^{-11}$ | $2.45 \times 10^{-11}$ | $1.22 \times 10^{-11}$ |
| | | Mean | | $7.32 \times 10^{-12}$ | $1.08 \times 10^{-11}$ | $1.39 \times 10^{-11}$ | $2.28 \times 10^{-11}$ | $1.95 \times 10^{-11}$ |
| | | Standard deviation | | $2.78 \times 10^{-12}$ | $7.64 \times 10^{-13}$ | $1.48 \times 10^{-12}$ | $2.89 \times 10^{-12}$ | $6.99 \times 10^{-12}$ |
| $\alpha$ | HTO | Asymptote | 5 | 0.5556 | 0.4815 | 0.4074 | 0.3333 | 0.2593 |
| | | Hydrus-1D | 5 | 0.5475 | 0.5327 | 0.4013 | 0.3733 | 0.3310 |
| | | Lsqcurvefit | 5 | 0.5722 | 0.5019 | 0.4312 | 0.3527 | 0.2787 |
| | | Mean | | 0.5584 | 0.5054 | 0.4133 | 0.3531 | 0.2897 |
| | | Standard deviation | | 0.0126 | 0.0258 | 0.0158 | 0.0200 | 0.0371 |
| | Tc | Asymptote | 5 | 0.3900 | 0.1700 | 0.0876 | 0.0433 | 0.0351 |
| | | Hydrus-1D | 5 | 0.3924 | 0.1692 | 0.0876 | 0.0433 | 0.0351 |
| | | Lsqcurvefit | 5 | 0.4000 | 0.2403 | 0.1370 | 0.1290 | 0.1000 |
| | | Mean | | 0.3941 | 0.1932 | 0.1041 | 0.0719 | 0.0567 |
| | | Standard deviation | | 0.0052 | 0.0408 | 0.0285 | 0.0495 | 0.0375 |

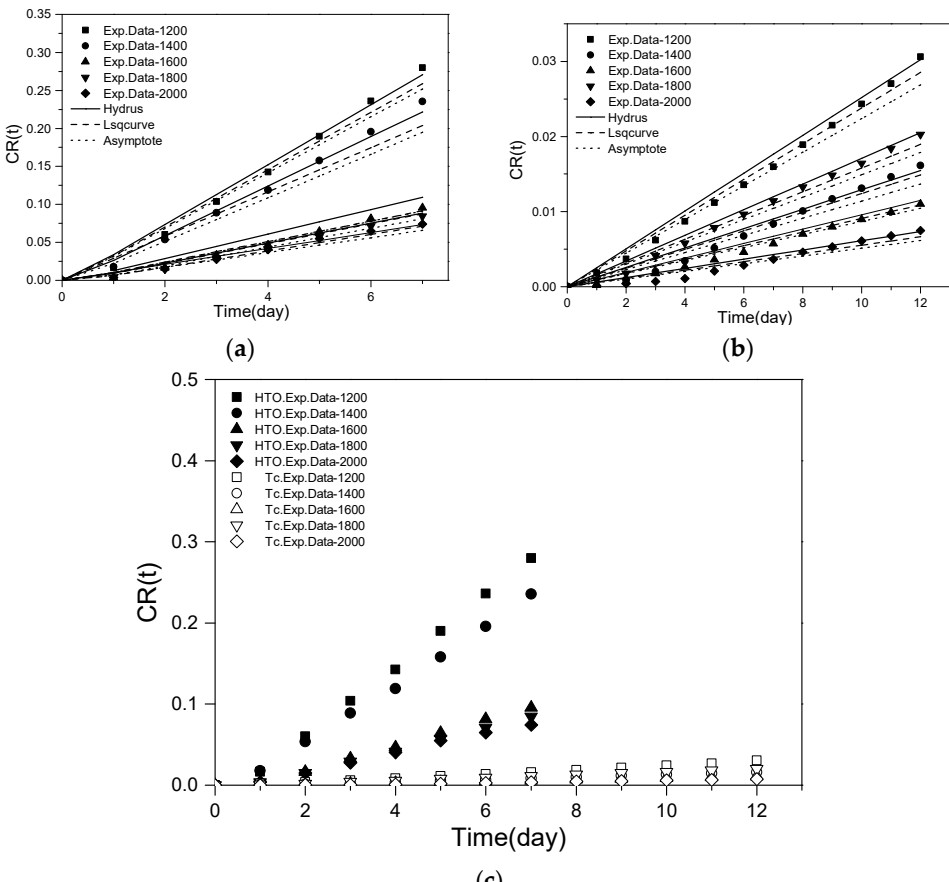

**Figure 3.** Various estimations of the comparable diffusion of HTO and Tc in bentonite compacted to various dry densities (1200, 1400, 1600, 1800, and 2000 kg/m³) (**a**) HTO; (**b**) Tc; (**c**) HTO and Tc.

### 3.3. Statistical Analysis of the Numerical Estimations of HTO and $^{99}$Tc

All numerical estimations of HTO and Tc at different densities had a high linear relationship ($R^2 > 0.99$), as shown in Figure 3 and Table 1, which lists the mean Da values of HTO and Tc ranging from 0.75 to $1.12 \times 10^{-10}$ m$^2$/s and 0.73 to $2.28 \times 10^{-11}$ m$^2$/s, respectively. Both HTO and Tc appeared to exhibit less diffusion behavior with increases in the compacted density due to changes in the pore structure of the bentonite (e.g., changes in constrictivity and tortuosity).

In fact, the apparent diffusion coefficient Da and the capacity factor $\alpha$ for $^{99}$TcO$_{4-}$ shown in Table 1 and Figure 4 were noticeably lower than those of HTO at all of the various densities as well as in the numerical estimations due to the larger radius of the hydrated anions (~ 3.5Å for $^{99}$TcO$_{4-}$) [6]. In addition, compared to HTO, the larger diameter of the $^{99}$Tc molecule and the strong anionic exclusion effects inhibiting the negative charge on the surface of the clay may account for these low diffusion coefficients [38,39]. In order to determine the factors and characteristics of HTO and Tc diffusion in compacted bentonite, significant differences ($p < 0.05$) using a one-way analysis of variance (ANOVA) are marked in Table 2 for the different numerical estimations, densities, and radionuclides (HTO and Tc). According to the $p$ value for the apparent diffusion coefficient (Da) and the capacity factor ($\alpha$) obtained using the SPSS analysis, there was no influence on the different numerical estimations.

**Table 2.** Results of the statistical analysis of the influence of different methods and different parameters for HTO and Tc using $p$ values.

| RN | Method of Estimation | Da | Density | $\alpha$ |
|----|----------------------|----|---------|----------|
| HTO | Asymptote | $6.40 \times 10^{-11}$ | 1200~2000 | 0.4074 |
|  | Hydrus-1D | $9.56 \times 10^{-11}$ | 1200~2000 | 0.4372 |
|  | Lsqcurvefit | $8.72 \times 10^{-11}$ | 1200~2000 | 0.4273 |
| Tc | Asymptote | $1.37 \times 10^{-11}$ | 1200~2000 | 0.1452 |
|  | Hydrus-1D | $1.66 \times 10^{-11}$ | 1200~2000 | 0.1455 |
|  | Lsqcurvefit | $1.43 \times 10^{-11}$ | 1200~2000 | 0.2013 |

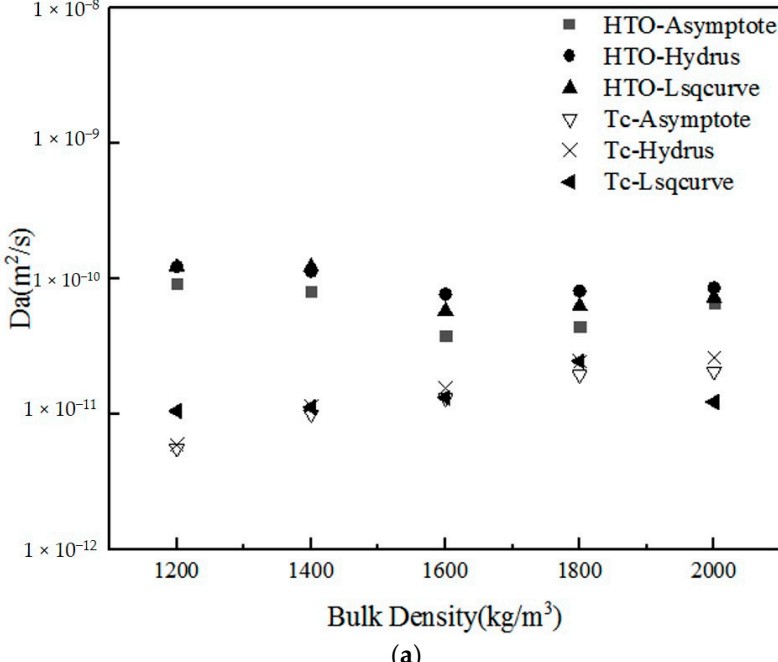

(a)

**Figure 4.** *Cont.*

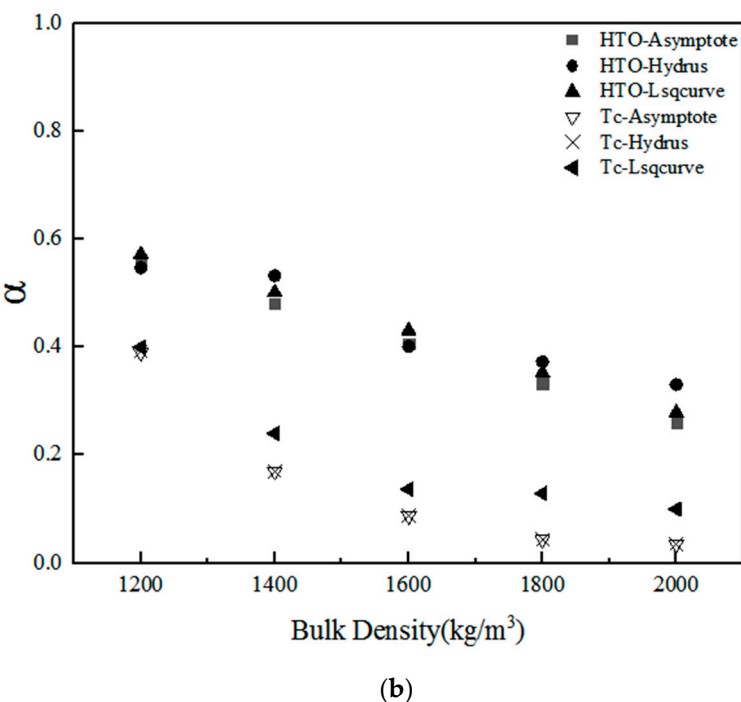

(**b**)

**Figure 4.** Different estimations of the diffusion parameters of HTO and Tc (**a**) apparent diffusion coefficient (Da); (**b**) capacity factor ($\alpha$).

## 4. Discussion

The FT-IR spectrum was used for the identification of clay minerals (especially montmorillonite with strong sorption capacity), poorly crystalline mineral phases, and possible adsorbed elements or functional groups. The results of the FT-IR, $N_2$-BET, XRF, and XRD analyses for MX-80 bentonite were in agreement with the commercial list and previous works, which justified a high level of confidence related to the diffusion experiments. Through three numerical analyses, including the graphical asymptote, the use of Lsqcurve-fit, a set of parameters for the apparent diffusion coefficient (Da) and the capacity factor ($\alpha$) were determined, with which the through-diffusion experimental results of HTO and Tc(IV) could be described adequately. Both HTO and Tc appeared to exhibit less diffusion behavior with increases in the compacted density due to changes in the pore structure of the bentonite (e.g., changes in constrictivity and tortuosity). In fact, the apparent diffusion coefficient Da and capacity factor $\alpha$ for Tc(IV) were noticeably lower than those of HTO at all of the various densities as well as in the numerical estimations due to the larger radius of the hydrated anions. Moreover, a SPSS statistical analysis (ANOVA) showed that obvious influences or significant differences were found for the various densities and radionuclides. These results also showed good agreement with those of previous studies [11,12,14,40,41], where the obvious differences in the diffusion parameters derived in the TD experiments could only be attributed to test conditions such as the cation or anion, the oxi-reducing conditions, the ionic strength, the pH in the aqueous phase, and differences in the type of geological materials employed.

## 5. Conclusions

In deep geological repositories (typically at a depth of approximately 500 m) for SNF or HLW, the environmental conditions responsible for chemical modifications will vary over time, from the initial excavations to the restoration of ambient geological conditions. The diffusion parameters obtained from the different estimation methods, such as the graphical asymptote, analytical solutions, and HYDRUS-1D used in the through-diffusion experiments may cause data uncertainties. It is thus a good alternative to acquire the diffusion parameters using a statistical analysis, which also provides a higher level of confidence

related to safety assessments of radioactive waste disposal in the future. According to results of the *p* value for the apparent diffusion coefficient (Da) and the capacity factor (*α*), there was no influence on the results of different numerical estimations. However, there were obvious influences or significant differences resulting from the different densities and radionuclides. These findings indicate that diffusion through the compacted bentonite samples was strongly affected by the pore structure. The apparent diffusion coefficient (Da) and the capacity factor (*α*) obtained in this study could be used as input data for safety assessments. Our results confirm that MX-80 bentonite is a potential candidate for a buffer/backfill material to counter the diffusion of radionuclides from high radwaste containers.

**Author Contributions:** Conceptualization, S.-C.T., C.-P.L., Y.H. and N.-C.T.; methodology, C.-P.L., D.C. and N.-C.T.; software, C.-P.L. and N.-C.T.; validation, Y.S., D.C., N.-C.T. and S.-C.T.; formal analysis, Y.S., D.C., C.-P.L. and N.-C.T.; investigation, Y.H., Y.S., I.-H.L. and C.-F.N.; resources, N.-C.T. and S.-C.T.; data curation, D.C., Y.H. and Y.S.; writing—original draft preparation, C.-P.L. and N.-C.T.; writing—review and editing, C.-P.L. and N.-C.T.; visualization, Y.H., Y.S., I.-H.L. and C.-F.N.; supervision, C.-P.L. and N.-C.T.; project administration, N.-C.T. and S.-C.T.; funding acquisition, N.-C.T. and S.-C.T. All authors have read and agreed to the published version of the manuscript.

**Funding:** This project was mainly supported by the Doctor Initial Financial Project (No. 1410000434), East China University of Technology. This project was financed in part by the Ministry of Science and Technology (MOST, Taiwan R.O.C.) and the Atomic Energy Council (AEC, Taiwan R.O.C.) through a 2-year mutual fund program project under contract numbers 109-2623-E-007-006-NU and 110-2623-E-007-004-NU.

**Acknowledgments:** This study was supported in the experimental part by Tsuey-Lin Tsai, Yu-Hung Shih and Liang-Cheng Chen, members of the Chemistry Division, Institute of Nuclear Energy Research Nuclear Science and Technology, Taiwan (ROC). The experimental and instrumental analysis of this study was supported by the Department of Nuclear Environmental Science, the China Institute for Radiation Protection (CIRP), Taiyuan, CHINA, the Instrumentation Center at National Tsing Hua University, and the National Synchrotron Radiation Research Center (NSRRC) in Taiwan under contract number 2020-1-123-5.

**Conflicts of Interest:** The authors declare no conflict of interest.

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
