# Peer review of "A Statistical Evaluation to Compare and Analyze Estimations of the Diffusion Coefficient of Pertechnetate (99TcO4) in Compacted Bentonite"

_minerals, doi:10.3390/min11101075_

Round 1

Reviewer 1 Report

The paper is written clearly and the topic fits well to the special issue. As an experimental clay scientist I was not able to assess the numerical modelling. Therefore I focused on the clay scientific part. Overall minor revision is required.

It is a pity that the authors did not provide line numbering. Therefore I have to explain which parts would need some improvement.

P2last sentence of 1st paragraph: what do the exchangeable cations buffer?

P2 2nd paragraph: at the end a dot “.” (or fullstop) is missing.

Ch2.1, third line: 1000 m ???

Same line: what is a “true” density? Is it particle density? Explain or provide reference.

Ch2.2, 5th line: why did XRF provide “semi-quantiative” data??? It is a quite accurate method if calibration is done properly.

CH2.2, 11th line: delete “in” before compared

Ch2.3, 3rd line: add space before number.

Ch2.4 last line: “-3” superscript

Ch2.4.2, 3rd line: either “s” behind “time” or “was” instead of “were”

Ch2.4.2, last sentence: this sentence would make sense if you would add “be” before “effective”, but is this what you ment?

Ch3.1, 3rd line: if at all there are only few SiOH groups in smectites. In the figure this band was assigned correctly. Text and figure should provide consistent information.

Ch3.1, 4th line: “2” subscript

Ch3.1, 5th line: this band is rarely assigned to an overtone, it is commonly assigned as H2O deformation. If you insist on your assignment you would need to add a reference.

Page6, 4th line: “2” subscript

P6, 6th line: delete “corresponding to …silicate” (that does not fit to the Si-O-Si bending mentioned before)

P6, 2nd paragraph, line3: “2” superscript

One line below: XRF data is given with 2 digits which does not correspond to the accuracy of the method (commonly 1 digit)

P6, 2nd paragraph, line 8: in a preceeding sentence the pyrite content is given and one sentence later it is stated that “no pyrite was found”. That has to be resolved.

Fig 2: the term “crystal absorption” is strange, use the same as in the text. Also the band assignement “MgAlOH” is wrong, this is the quartz doublet.

Ch3.2, 13th line: “-3” superscript

P7, 3rd line: this sentence does not seem to be finished (not only because a dot is missing)

Table 1: 1st line: “3” superscript

Table 2 + Fig3: “dot” at the end of the captions.

Discussion, 1st sentence: there is a verb missing in this sentence.

Page10, after (ANOVA): sentence has to be improved

Conclusions, 10th line: there was no influence on (the results of) the numerical… or there was no influence of the different numerical estimations?

Author Response

Dear Reviewer

We are very grateful to you for all your kindly assistance, especially for our English typo , word correction and professional comments. It has been corrected and improved by asking native English speaker help us with proper knowledgeable expressions.

For FTIR analysis in our manuscripts , it could be explained that FTIR is a simple and quick composition analysis for scanning MX-80 bentonite by evaluating the different molecule and function groups. For previous works , several advanced analysis such as XANES, EXAFS, XPS etc.[1,2] were also applied to find the interaction between radionuclides and minerals surface complexation(or atoms). Therefore, there are several key and important function groups such as -OH, Si-O-Si , Al-O etc. in MX-80 bentonite by FTIR and it also would be a effective and good evidence to provide more information in diffusion behavior of HTO and Tc in our work.

  1. Liu,W.T.,Tsai, S.C., Tsai, T.L., Lee, C.P. , C.H.Lee. Characteristic study for the uranium and cesium sorption on bentonite by using XPS and XANES (2017) Journal of Radioanalytical and Nuclear Chemistry 314, 2237–2241
  2. Liu,W.T., Tsai, S.C., Tsai, T.L.,Lee, C.P. , Lee, C.H. (2019) An EXAFS study for characterizing the time-dependent adsorption of cesium on bentonite. Environmental Science: Processes & Impacts . 21, 930–937

Finally, we appreciate your kind and excellent suggestion for our manuscript, and Thanks a lot!

Best Regards

N-C Tien

Reviewer 2 Report

See attachment for both general and specific comments.

Author Response

Dear Reviewer

Thanks for your valuable comments and suggestion.

Firstly, we are sorry for the unclear expression and some superfluous typo in our manuscript. It has been corrected and improved by asking native English speaker help us with proper knowledgeable expressions

For statistics analysis in our manuscripts , it could be explained that we applied different numerical analysis by evaluating the diffusion concentration profiles with the root mean square errors. Moreover, a set of 5 columns is compacted with decreasing bulk density (No 1. ~5), and diffusion coefficients of HTO and Tc also showed a tendency to decrease with increasing bulk density. In fact, we applied TD columns for several years, and uncertainty could be estimated around 5 to 10% in our TD system according to our experience. A statistical evaluation using a one-way analysis of variance (p-value) could provide a more confidence in parameter estimation to reduce the experimental uncertainties in our diffusion results.

Therefore, the key and important issue for safety assessment(SA) in HLW disposal is the reliable experimental data (Lab and In-situ) for different testing conditions, and it is better to find a suitable and reliable method to estimate the parameter by different numerical analysis. Our object in this work is to compare the HTO and Tc diffusion behavior by 3 numerical analysis and we applied a ANOVA (p<0.05) to know there is no influence on different numerical results in Lab experiments. It also could be a good method to apply in-situ experimental data in following works.  

Finally, we appreciate your kind and excellent suggestion for our manuscript, and Thanks a lot!

Best Regards

N-C Tien

Round 2

Reviewer 2 Report

Dear Authors,

Please find the attached comments to clarify your results of statistical analysis.

Author Response

Dear Reviewer

Thanks for your valuable comments and suggestion. It looks good in Table 2 according to your suggestion, and we appreciate your kind and excellent suggestion for our manuscript, and Thanks a lot!

Best Regards

N-C Tien